# When Does Alzheimer′s Disease Really Start? The Role of Biomarkers

**DOI:** 10.3390/ijms20225536

**Published:** 2019-11-06

**Authors:** Ana Lloret, Daniel Esteve, Maria-Angeles Lloret, Ana Cervera-Ferri, Begoña Lopez, Mariana Nepomuceno, Paloma Monllor

**Affiliations:** 1Department of Physiology, Faculty of Medicine, University of Valencia, Health Research Institute INCLIVA, Avda. Blasco Ibanez, 17, 46010 Valencia, Spain; daniel.esteve@ext.uv.es (D.E.); mary.pinheiro@gmail.com (M.N.); paloma.monllor@uv.es (P.M.); 2Department of Clinic Neurophysiology. University Clinic Hospital of Valencia, Avda. Blasco Ibanez, 19, 46010 Valencia, Spain; malloretalc@gmail.com; 3Department of Human Anatomy and Embriology, Faculty of Medicine, University of Valencia, 46010 Valencia, Spain; ana.cervera-ferri@uv.es; 4Department of Neurology. University Clinic Hospital of Valencia, Avda. Blasco Ibanez, 19, 46010 Valencia, Spain; blpesquera@hotmail.com

**Keywords:** dementia, AD spectrum, biomarkers, CSF, AD dynamic, imaging biomarkers

## Abstract

While Alzheimer’s disease (AD) classical diagnostic criteria rely on clinical data from a stablished symptomatic disease, newer criteria aim to identify the disease in its earlier stages. For that, they incorporated the use of AD’s specific biomarkers to reach a diagnosis, including the identification of Aβ and tau depositions, glucose hypometabolism, and cerebral atrophy. These biomarkers created a new concept of the disease, in which AD’s main pathological processes have already taken place decades before we can clinically diagnose the first symptoms. Therefore, AD is now considered a dynamic disease with a gradual progression, and dementia is its final stage. With that in mind, new models were proposed, considering the orderly increment of biomarkers and the disease as a continuum, or the variable time needed for the disease’s progression. In 2011, the National Institute on Aging and the Alzheimer’s Association (NIA-AA) created separate diagnostic recommendations for each stage of the disease continuum—preclinical, mild cognitive impairment, and dementia. However, new scientific advances have led them to create a unifying research framework in 2018 that, although not intended for clinical use as of yet, is a step toward shifting the focus from the clinical symptoms to the biological alterations and toward changing the future diagnostic and treatment possibilities. This review aims to discuss the role of biomarkers in the onset of AD.

Alzheimer’s disease (AD) has always been a primarily clinical disease, seeing as its confirmation could only be reached through histopathological post-mortem studies. However, the more its physiopathology is known, the more certainty should be put into diagnosing it. Therefore, scientists have searched for biomarkers to help as diagnostic tools. Nevertheless, the fast rise of biomarkers gave rise to many questions such as, what biomarkers exist today? How can they be used in AD’s diagnosis? When does AD really start? In this review, we aim to answers those questions.

## 1. The Classical Diagnostic Criteria

Diagnosing AD has been an absolute challenge since it was described by Alzheimer at the beginning of the 20th century [1]. In the last part of this same century, the first diagnostic criteria were established by the United States National Institute for Communicative Disorders and Stroke—the Alzheimer’s Disease and Related Disorders Association (NINCDS-ADRDA) [2] and by the Diagnostic and Statistical Manual of Mental Disorders of the American Psychiatric Association (DSM-IV) [3].

The NINCDS-ADRDA-DSM-IV criteria defines AD as a syndrome, and its diagnosis has the following three grades of certainty: probable, possible, and definite AD, the latter usually requiring a post-mortem histopathological confirmation to reach diagnostic certainty. The sensitivity of the NINCDS-ADRDA-DSM-IV diagnostic criteria varied between 65–96% [4,5,6,7] and their specificity between 23–88% [6,7] because other dementias such as Lewy bodies dementia, frontotemporal dementia, and vascular dementia could not be completely excluded [6,8].

## 2. The Inclusion of Biomarkers in the Diagnostic Criteria

As the knowledge on the pathophysiological, molecular and structural changes in AD increased, Dubois et al. [8] revised AD’s classical diagnostic criteria and proposed new ones. These new diagnostic criteria aimed to identify AD in its earlier stages, before the development of a dementia syndrome. They established a specific clinical phenotype of AD, casting aside the diagnosis of exclusion and solving the problem of low diagnostic specificity, while also offering the chance of an early therapeutic intervention [8]. Their principal criterion is a failure in episodic memory that appears early in the disease (A, see Table 1). Furthermore, they introduced as a novelty the support criteria which are based on biomarkers (B, C, D, and E). Therefore, the presence of the main criterion together with at least one of the supporting criteria is indicative of AD pathology.

With these criteria, Dubois et al. [8] arrive at the diagnosis of AD earlier and in a more specific way than their predecessor, the NINCDS-ADRDA criteria, covering the earliest stages of the disease, when there is still no dementia syndrome, as well as the later stages when the patient is already functionally disabled. The ingenuity resides on the fact that biological biomarkers are included in the criteria for the first time, as a requirement to reach diagnosis. These biomarkers include structural and molecular imaging, cerebrospinal fluid (CSF) analysis, and genetic mutation analysis. 

After Dubois’ first step in including biomarkers as diagnostic criteria, in 2011, the National Institute on Aging and the Alzheimer’s Association (NIA-AA) defined new diagnostic criteria that separated the disease in three clinical stages, each one with its own diagnostic recommendation. First is the preclinical stage that presents pathologic brain changes, which may be in progress decades prior to disease, without evident clinical symptoms. In this stage, alterations can be seen in CSF and imaging biomarkers although, at present, they cannot predict which of these individuals will develop dementia [9]. The second stage is Mild Cognitive Impairment (MCI), which is marked by memory symptoms that are greater than normal for a person’s age and education but that do not interfere with their independence and may or may not progress to Alzheimer’s dementia [10]. The final stage is Alzheimer’s dementia, in which symptoms are significant enough to impair a person’s ability to function independently [11]. 

The definition of a preclinical stage that could be in progress many years before the beginning of symptoms meant that the biomarkers acquired a greater importance in the diagnostic process.

## 3. Alzheimer’s Disease Biomarkers

Biomarkers are defined as physiological, biochemical, or anatomical variables that can be measured in vivo and that characterize specific pathological changes of a disease. We can classify AD biomarkers, according to the method of analysis, into biochemical CSF biomarkers or imaging-derived biomarkers [12,13].

### 3.1. Biomarkers in Cerebrospinal Fluid

CSF biomarkers are, nowadays, widely used in routine clinical practice to support the diagnosis of MCI and AD [14,15,16]. The levels of beta-amyloid peptide (Aβ), total tau (t-tau), and phospho-tau (p-tau) in CSF are used as specific biomarkers of AD, since they reflect the pathologic processes of Aβ42 aggregation and tau’s hyperphosphorylation, respectively, and are included within the support criteria for the clinical diagnosis of probable AD [8].

#### Aβ42 and Tau as Biomarkers

There is an inverse correlation between brain amyloid load and CSF Aβ42 levels, with the latter being diminished in AD patients with respect to healthy subjects. The levels of Aβ in CSF reflect the pathologic process of aggregation of this peptide into amyloid plaques. Thus, a reduction on its clearance toward the CSF happens, and its concentration decreases. In early stages of AD, Aβ levels are already altered in the CSF; in fact, there are studies evidencing that CSF Aβ begins to show abnormal levels several years before the appearance of the first subjective memory complaints [17], which makes it the earliest marker that exists today [18].

On the other hand, tau levels in CSF are a reflection of tau’s pathogenesis in the cerebral cortex [19]. Although p-tau should be a more specific indicator of AD than t-tau, both t-tau [20] and p-tau [20,21] behave in a very similar way in AD, increasing their concentrations in CSF. Both increments are associated with the load of neurofibrillary tangles and are indicators of neuronal damage [13,21,22,23]. Although an increase in CSF tau is not specific to AD, it does correlate with the clinical severity of the disease, increasing its levels at the same time as cognitive failure increases [22]. 

In early stages of AD, lower levels of Aβ appear in CSF, and it is considered a predictor of the evolution of MCI to AD [23] and, in the same way, high levels of p-tau and t-tau in CSF can predict with good accuracy an incipient AD in patients with MCI [24]. These parameter alterations appear even in cognitively normal subjects, where it is possible to detect abnormalities of Aβ and tau in CSF many years before MCI is diagnosed [25,26,27,28,29,30,31,32,33].

Interestingly, it has been shown that cognitively normal carriers of the Apolipoprotein ε4 allele (APOE4), who are at risk for late-onset AD, also show decreased levels of Aβ in CSF [33,34]. In familial AD, studies have seen that asymptomatic carriers of PSEN1 and APP mutations show alterations in the levels of CSF Aβ and p-tau that precede the clinical onset of the disease by more than 10 years [35,36,37].

### 3.2. Imaging Biomarkers in AD

#### 3.2.1. PiB-PET

The Pittsburgh compound B (PiB) is a specific ligand of Aβ that, when used as a tracer in a positron emission tomography (PiB-PET), makes it possible to analyze in vivo both cerebral Aβ load and Aβ spatial distribution. It has been shown that ante-mortem imaging studies with PiB-PET represent a direct measurement of amyloid plaque burden [38,39,40,41,42,43,44] and correlate well with post-mortem studies [44,45]. Furthermore, the cerebral load of PiB is inversely related to the levels of CSF Aβ [24,27,46].

The conversion of a subject from PiB-negative to PiB-positive occurs at a very early stage of the development of AD [47]. When considering cognitive normal subjects, APOE ε4 allele carriers have a higher rate of conversion to PiB-positive that happens years before the clinical onset of AD than those that do not carry the gene. Moreover, asymptomatic carriers of PSEN1 and APP mutations present a higher load of PiB in the cortex and striatum [48,49,50]. Nevertheless, there are also cognitive normal elderly individuals that do not carry genetic risk factors that present a positive PiB signal. Therefore, PiB-PET is only recommended when other clinical evidences of dementia are present [19,26], even though it is considered a valid marker to help in the diagnosis [51].

Recently, the advent of tau-specific ligands for PET imaging allows the usage of this technique as a biomarker [52,53,54]. It could be thought that this new Tau-PET might compete with the stablished amyloid-PET, but a recent work proposes that combining both could be better to track AD progression [55,56]. 

#### 3.2.2. FDG-PET

PET imaging with 2-deoxy-2 [18F] fluoro-D-glucose tracer (FDG-PET) measures the cerebral metabolism of glucose and is an indicator of neuronal and glial function. In AD, FDG-PET signal decreases, which is consistent with glucose hypometabolism and synaptic dysfunction, and it also has a specific topographic distribution pattern [57]. In addition, it correlates with decreased levels of synaptophysin found post-mortem that indicate a loss of synaptic activity [58,59]. Furthermore, a bilateral reduction in FDG uptake in the temporal and parietal regions and especially in the cingular cortex is described in individuals with AD [60,61]. Moreover, FDG uptake is inversely related to the cognitive deficit, that is to say, less uptake correlates with greater cognitive damage throughout the entire clinical spectrum of the disease [62] with a sensitivity and a specificity greater than 80% [63,64]. Cognitively normal APOE4 carriers also have glucose hypometabolism [65,66], which is worse in homozygotes than in heterozygotes [67], and can be detected as early as the third decade of life [68]. 

Finally, it is now possible to assess the regional distribution and the total tau burden in vivo with new PET imaging radiotracers. In this regard, it has been shown that tau-PET is a more sensitive marker for the detection of the earliest cognitive changes in AD than Aβ PET and cortical thickness measures [69].

#### 3.2.3. Structural and Functional Magnetic Resonance Imaging (MRI) 

Cerebral atrophy, specifically in mesial-temporal structures, can be quantified using structural magnetic resonance imaging (MRl) [60,70] and can be detected before the appearance of the first clinical symptoms. In fact, many studies defend this biomarker as a reliable diagnostic tool [71] and as a neurodegeneration marker [61,72]. Indeed, it is included in both the Dubois [8] and the NIA-AA [16] diagnostic criteria and its specificity and sensitivity as an AD marker is greater than 85% [60].

In early AD, the first detectable signs that can be observed in a structural MRI are atrophy in the middle temporal lobe (affecting specially the hippocampus), and a decrease in the thickness of the cerebral cortex in regions that are vulnerable to AD [73,74,75,76,77,78,79,80,81]. In asymptomatic carriers of APP mutations, a decrease in hippocampal volume can be identified 2–3 years before the onset of dementia [82] and, in elderly people, this alteration can be detected up to six years before [72,73,74,75,76,77]. Moreover, in cognitively normal elders, CA1 region abnormalities represent an early predictor of the development of dementia [74]. In addition, the loss of volume in the entorhinal cortex precedes cognitive decline by four years and has a predictive power of up to 90% [75].

## 4. When Does AD Really Start?

With the advent of biomarkers came a conceptual change of the disease. We moved from a “static and defensive” view of the pathogenesis of AD to a “dynamic and compensatory” point of view. In the first viewpoint, the brain lesions that lead to neuronal and synaptic loss and finally to cognitive deterioration depend on the degree of external aggression and on the structural reserve that each person has. The current view considers an inter-individual variability in the response to these initial aggressions, as well as differences in the severity of the pathological process and in the efficiency and evolution over time of the cerebral compensatory mechanisms [83,84,85]. Therefore, the idea that AD’s main pathological processes have already taken place before we can clinically diagnose MCI has been established, and this is reinforced by the fact that these lesions begin even decades before the appearance of the earliest symptoms, when the subject is still cognitively normal [86]. Therefore, this change in perspective increasingly supports the need for early therapeutic action in order to compensate for those biological processes that are already compromised before the onset of the cognitive failure [87].

AD is now considered a neurodegenerative disease with a very long evolution that starts silently decades before the onset of symptoms and advances gradually and slowly until it compromises the person′s cognition. Therefore, we moved from a static vision of AD in which a person is affected or not by the disease, to a dynamic concept of AD, in which dementia is considered the final stage of a set of pathological changes that occur in a chronic and gradual manner. 

In this progression, which may take years, biomarkers can anticipate the clinical manifestations of dementia and, as the new diagnostic criteria introduced biomarkers in a supporting role in AD’s diagnosis, many laboratories worldwide have already started using them. With this in mind, and based on determinations made in different populations, Jack and collaborators proposed a model for the evolution of AD over time, known as “the dynamic biomarker cascade model” [16]. In this model, biomarkers do not increase all at once but do so in an orderly manner, a concept that is reinforced in the work by Dubois and collaborators [18]. The model presents three phases along the continuum of the disease—first, the cognitively normal asymptomatic phase, then the MCI phase that begins to show clinical affectation, and finally the dementia phase. All over this spectrum, the biomarkers described previously would present abnormal levels as the disease evolves and, eventually, correlate with the clinical symptoms presented by the patients. 

Jack and collaborators propose as the initial event the abnormally increased levels of Aβ that would lead to the formation of cerebral amyloid plaques. This would be reflected in the decreased levels of Aβ in CSF and in the increased amyloid load in PiB-PET, and these alterations would appear while the individuals are still cognitively normal. Afterward, there would be an increase in CSF tau abnormalities followed by alterations in FDG-PET. These are biomarkers of neuronal dysfunction and neurodegeneration and correlate with the severity of clinical symptoms. Lastly, in advanced stages, structural brain changes would appear such as cortical atrophy and decreased hippocampal volume that could be detected by MRI [16].

A very recent work by Petrella and collaborators [88] developed a mathematical causal model of the dynamic biomarker cascade theory in AD, which might help to explain how these biomarkers interact and evolve over time and could potentially help patients, researchers, and medical personnel. This is a great advancement in the knowledge of the disease, but there is still a long way to go. Although, biomarkers could have a role in predicting whether a patient could convert from MCI to AD, there is not a consensus on which biomarkers could assume that role [89,90].

However, the scientific community’s efforts go beyond designing computational models to determine the behaviour of different biomarkers in the evolution of the disease. Models have been designed for many different aspects of the disease, such as a model based on the amyloid cascade hypothesis, showing the effects of pathological processes such as oxidative stress, inflammation or cerebrovascular disease in the kinetic of Aβ aggregation [91]. Moreover, another model focused on synaptic loss and compensation by the reinforcement of the remaining connections [92] and, more recently, Ding et al. (2018) designed a hybrid computational approach for a more accurate disease severity classification [93].

Nevertheless, as scientists started to better understand AD’s pathophysiology, the biggest challenge became designing computational models capable of predicting the efficacy of a specific treatment. To reach this objective, models have been created analysing potential treatments. Anastasio (2013) incorporated the role of estrogens in Aβ regulation into a model that can generate therapeutic predictions and the possible benefits of this therapy [94]. This model showed that estrogen could reduce Aβ and that non-steroidal anti-inflammatory drugs could provide a small additional benefit. 

Furthermore, immunotherapy, probably the most promising treatment for AD at this moment, was also analysed by computational models. Diem et al. (2016), have incorporated this therapy’s possible complications into their model and concluded that a failure in periarterial drainage seems to be an important mechanism [95]. Another computer simulation model pointed out that immunotherapy against Aβ might not be effective, unless it is used during early stages of AD [96] or combined with other therapies. However, a more recent model simulated the differential impact of Aβ oligomers on glutamate and nicotinic neurotransmission while under different treatments, including a passive vaccination with the monoclonal antibody solanezumab, the use of the beta-secretase inhibitor verubecestat, and of the gamma-secretase inhibitor semagacestat. They predicted a cognitive worsening in people with low Aβ baseline and an improvement in those with moderate to high Aβ levels [97]. 

Computational models analyzing neurotransmitters have also been created. One such model has been implemented using preclinical data available on receptor pharmacology of cholinergic and catecholamine neurotransmitters and clinical data, to predict the effects of memantine, an N-Methyl-D-aspartic acid (NMDA) inhibitor, in different phases of AD pathology [98]. 

Finaly, Stefanovski et al. (2019) created a computational multi-scale brain model, using the Virtual Brain Platform, and including PET and electroencephalogram, to simulate regional neural activity and hyperexcitability in AD and how it relates to Aβ. This model reveals a potential functional reversibility of large-scale alterations in AD after memantine treatment [99].

## 5. Abnormalities in Biomarkers Precede Clinical Symptoms

In the “dynamic biomarker cascade model,” each biomarker reaches its maximum effect at a certain moment in the progression of the disease, and that happens in an orderly manner over time. Interestingly, the maximum levels can be detected in a person before any clinical symptom. In fact, several studies have shown that 20–40% of cognitively normal old present Aβ deposits in their cerebral tissue [22,100,101]. Moreover, in post-mortem samples from non-demented elderly people, Aβ plaques were also present [102,103]. Therefore, deposits of amyloid plaques alone, even in significant quantities, are not enough to produce dementia [16,26,27,30]. Not only Aβ but also tangles may be present in subjects without cognitive decline. Nonetheless, in asymptomatic patients, the presence of neurofibrillary tangles tends to be limited to the entorhinal cortex (stages of Braak I–II), while in symptomatic subjects, tangles are much more widespread [102,103]. 

On the other hand, data obtained through imaging studies with PiB-PET suggests that Aβ deposits may appear up to two decades before the onset of clinical manifestations of dementia [86]. This concurs with the fact that, after the diagnosis of AD, the levels of Aβ in CSF do not change significantly, since it has already reached a plateau, and also with the pattern of PiB retention, which is not modified throughout the disease’s spectrum [86]. Consistent with the previous studies, investigations on familiar AD have shown that these alterations in Aβ and p-tau precede by more than 10 years the clinical onset of the disease [104]. Very recently, sequential changes in normal older adults, from Aβ to tau to cognition, have been described using repeated tau-PET and amyloid-PET measures to detect the earliest AD pathologic changes [105].

These studies are all in line with a final concept that this model postulates—the existence of a latent phase of variable duration between plaque formation and the onset of the neurodegenerative cascade. This could be due to differences in the processing of Aβ, to the capacity of resistance to pathological damage derived from the toxicity of Aβ and to compensatory mechanisms [106].

There is an increasing idea that AD pathology would trigger cerebral compensatory mechanisms all across the AD spectrum, and it would be in the preclinical phase that these mechanisms would begin to appear. However, there is still no consensus regarding the role that compensatory mechanisms might play in cognitively healthy subjects at risk of AD with positive biomarkers, and also regarding the influence they could have on the conversion to dementia. Lazarczyk et al. [104] suggest that the compensatory mechanisms would be divided into two categories: the passive ones (matching the cognitive reserve concept) and the active ones. The former would delay conversion to dementia, and the latter could stop disease progression in the preclinical phase and effectively prevent conversion to dementia.

Anatomically, this compensatory mechanism can be seen in structural changes found in asymptomatic people carrying the presenilin 1 mutation when compared to age-matched controls [107]. These individuals present a cortical thickening mainly in temporal and parietal areas, extending into precentral and postcentral cortex and pars triangularis, and also in structures of the posterior midline, such as precuneus and posterior cingulate. No areas of cortical thinning are observed in asymptomatic carriers, unlike those found in symptomatic people. Furthermore, structural changes are not limited to subjects with mutations in AD determinant genes, since they are also seen in the sporadic form of the disease. Some studies have shown that healthy subjects with evidence of initial deposits of Aβ [108] present greater volume and thickness in AD related cortical regions. With AD progression, these areas suffer a progressive thinning of gray and also white matter reaching the atrophy observed in more extensive regions in the symptomatic stage [109]. 

It would be interesting to have more studies evaluating these compensatory mechanisms, to understand it better and, if possible, to add it to the current computational models evaluating AD’s pathology and possible treatments, as they seem to be present since the very early stages of the disease. 

## 6. Influence of Risk Factors in Biomarker Dynamics

In 2013, Jack and Holtzman proposed a new model [17] where the subjects are classified by the time needed by a subject to progress through the entire spectrum of the disease and not by clinical stage. However, this time varies for each individual. According to this model, people with high risk of developing the disease would show greater abnormality in the biomarkers when compared to those individuals with low risk at the same age. High-risk subjects would be those carrying genetic allele mutations such as APP/PSEN1, as well as those with low cognitive reserve and those whose lifestyles increase the probability of cognitive damage. Low-risk subjects with a protective genetic profile, a high cognitive reserve, no pathological brain comorbidities, and a low-risk lifestyle for dementia could maintain a normal cognitive function despite developing AD pathology (Figure 1). Interestingly, Aβ-positive patients with greater cognitive reserve show attenuated clinical progression in pre-dementia stages of AD but accelerated cognitive decline after the onset of dementia [110]. 

Recently, McDade et al. [111] carried out a longitudinal study with high-risk subjects carrying autosomal dominant mutations and showed that the sequence and temporal dynamics of the biomarkers match the theoretical model. Another very recent study with more than 3600 participants showed two patterns of changes in biomarkers, one in APOE4 carriers and another in non-carriers. Subjects without the APOE4 allele showed an initial increase and then a decrease in CSF Aβ42 with a progression of CSF tau, while subjects with at least one APOE4 allele showed only a decrease in CSF Aβ42 associated with a progression of tau pathology, and both markers became positive with the progression of the disease [112].

## 7. New AD Classification Based on Biomarkers

Biomarkers provide a powerful tool for the clinical practice as well as for research, especially in human clinical trials, by improving diagnosis. This knowledge leads to an increase in clinical trials evaluating possible drugs that could cure, ease or control AD, but the efforts are still challenging. 

Since the gap between the pathological processes and the cognitive symptoms has proven itself quite large, and the risk of generating hypotheses that were not based on the pathophysiological changes of AD was a reality, the NIA-AA [113] proposed a new framework for the use of biomarkers in observational and interventional research. Therefore, the A/T/N system was described classifying subjects according to the number of positive biomarkers they presented. This system includes a binary system (positive or negative) depending on the measured biomarker. “A” refers to the value of a β-amyloid biomarker (amyloid PET or CSF Aβ42), “T” to the value of a tau biomarker (CSF p-tau, or tau PET), and “N” to biomarkers of neurodegeneration or neuronal injury (18FDG–PET, structural MRI, or CSF total-tau) [114]. A person with a positive “A” biomarker is classified as being in the “Alzheimer’s continuum”, that denotes either Alzheimer’s pathologic changes or AD, and those with positive biomarkers in both “A” and “T” categories are classified as having AD [113]. The possible outcomes of this classification system are illustrated in Table 2.

A recent study analysing the prevalence of biologically defined AD compared to clinically defined probable AD reports that the former is more prevalent than the latter, especially at age 85 years, a difference that is mostly driven by asymptomatic individuals with biological Alzheimer disease, that could be diagnosed after the A/T/N description [115]. 

Therefore, this research framework defines AD by its pathological alterations that could be documented by biomarkers and not by its clinical consequences. Although, at the moment, it is intended only for research and not routine clinical care [113,116], this new viewpoint might improve the selection of subjects and the conception of new therapies for clinical trials.

## 8. Conclusions

As we have seen in this article, our understanding of AD’s pathophysiology has changed considerably over the last few years. The disease is now considered a continuum, were the first stage begins decades before clinical symptoms are evident. This stage can be detected only with the help of specific biomarkers that identify the pathological process that is already in progress. Different models of the biomarkers’ progress have been created that consider their presence, the time they take to rise, their correlation to the individual’s cognitive status, and other factors that may contribute to an individual’s cognitive resilience despite the pathologic load. The last advance in this field was the new research framework defined by the NIA-AA in 2018. Although not indicated for use in clinical settings, this framework unifies all the biomarkers and their use and identifies them as the most important diagnostic factors. This vision ultimately changes the idea of AD as a clinical disease into a biological disease that, in reality, begins decades before any symptom start. This creates many new future possibilities, not only in the search for a more accurate and early diagnosis, but also in the search for more specific and efficient treatments. 

## Figures and Tables

**Figure 1 ijms-20-05536-f001:**
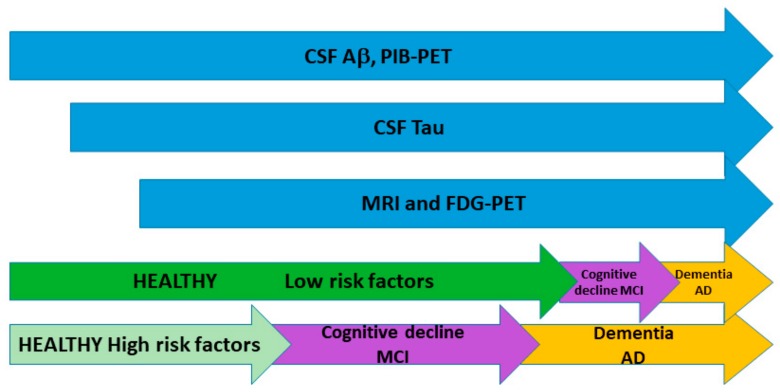
Evolution of the different AD biomarkers along time. Cerebrospinal fluid (CSF) Aβ and Pittsburgh compound B positron emission tomography (PiB-PET) abnormalities appear first followed by an increase in CSF tau levels and finally by structural alterations evidenced by magnetic resonance imaging (MRI) and fluoro-d-glucose tracer positron emission tomography (FDG-PET) (blue). The presence of risk factor determines the onset of the disease. People with low risk factors will present cognitive decline later than those with high risk factor (green).

**Table 1 ijms-20-05536-t001:** Main proposed criteria for Alzheimer’s disease (AD) diagnosis by Dubois et al. (2007) [8].

Criteria	Options
Main Criteria for AD Diagnosis (Obligatory)	(A) Early episodic memory failure represented by a gradual or progressive memory dysfunction at the beginning of the disease, informed by the patient or family, lasting more than six months. Associated with objective evidence of significant decline in episodic memory through tests (deferred memory).
Support Criteria for AD Diagnosis (At least one present)	(B) Loss of volume of the hippocampus, entorhinal cortex, amygdala or other mesial-temporal structures, evidenced by magnetic resonance imaging (MRI).
	(C) Abnormality in CSF biomarkers such as - Low concentrations of Aβ; - Increased t-tau or p-tau concentrations; or - A combination of the three.
	(D) Specific metabolic pattern evidenced by PET such as hypometabolism of glucose in bilateral temporal parietal regions.
	(E) Autosomal dominant family genetic mutations On chromosomes 21 (APP), 14 (PS1), or 1 (PS2).

**Table 2 ijms-20-05536-t002:** Outcomes on the biomarker’s A/T/N classification [113]. A: amyloid. N: Neurodegeneration. T: Tau.

A/T/N Profiles			Biomarker Outcome	Diagnosis
A+	T+	N+	AD	AD SPECTRUM
		N−	AD	
	T−	N+	AD and suspected non-AD pathologic change	
		N−	Alzheimer’s pathological change	
A−	T+	N+	Non AD pathological change	No AD
		N−	Non AD pathological change	
	T−	N+	Non AD pathological change	
		N−	Normal BIOMARKERS	

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
