# Peer review of "When Does Alzheimer′s Disease Really Start? The Role of Biomarkers"

_ijms, 2019, doi:10.3390/ijms20225536_

Round 1

Reviewer 1 Report

Lloret et al, have very nicely organised and summarised the role of Biomarkers in Alzheimer's disease in this mini review. The authors have also highlighted the alterations that may be observed from the pre-clinical stages to MCI to then AD. The manuscript is written in an appropriate English form.

Author Response

We thank the reviewer for his/her kind comments.

Reviewer 2 Report

Very good manuscript overall it is well written and very informative for the scientific community in understanding AD and its diagnosis.  

I would recommend doing a brief introduction to the topic as the reader expected to read a research study.  This manuscript appears to be a literature review informing the reader of the role of biomarkers in the diagnosis of AD.  I would recommend not just citing or summarizing the literature, but discussing it critically, identifying potential methodological problems, and pointing out any evident research gaps.  

The reader had difficulty following Table 1, particularly correlating the main criteria with the supporting criteria.  It is not clearly laid out. 

Author Response

Very good manuscript overall it is well written and very informative for the scientific community in understanding AD and its diagnosis.  

We thank the reviewer for this positive feedback.

I would recommend doing a brief introduction to the topic as the reader expected to read a research study. 

We have added to the Abstract Section the next sentence:

“This review aims to discuss the role of biomarkers in the onset of AD”.

And we have also added the following paragraph at the beginning of the manuscript:

“Alzheimer’s disease (AD) has always been a primarily clinical disease, seen as its confirmation could only be reached through histopathological post-mortem studies. However, the more its physiopathology is known, the more certainty should be put into diagnosing it. Therefore, scientists have searched for biomarkers to help as diagnostic tools. Nevertheless, the fast rise of biomarkers gave way to many questions such as: What biomarkers exist today? How can they be used in AD’s diagnosis? When does AD really start? In this review we aim to answers those questions. “

This manuscript appears to be a literature review informing the reader of the role of biomarkers in the diagnosis of AD.  I would recommend not just citing or summarizing the literature, but discussing it critically, identifying potential methodological problems, and pointing out any evident research gaps. 

We thank the reviewer for this helpful comment. We have added a little discussion about the compensatory cerebral mechanisms that maybe is the clearest research gap:

“There is an increasing idea that AD pathology would trigger cerebral compensatory mechanisms all across the AD spectrum, and it would be in the preclinical phase that these mechanisms would begin to appear. However, there is still no consensus regarding the role that compensatory mechanisms might play in cognitively healthy subjects at risk of AD with positive biomarkers, and also regarding the influence they could have on the conversion to dementia. Lazarczyk et al (2012) suggest that the compensatory mechanisms would be divided into two categories: the passive ones (matching the cognitive reserve concept) and the active ones. The former would delay conversion to dementia; and the latter could stop disease progression in the preclinical phase and effectively prevent conversion to dementia.

Anatomically, this compensatory mechanism can be seen in structural changes found in asymptomatic people carrying the presenilin1 mutation when compared to age-matched controls. These individuals present a cortical thickening mainly in temporal and parietal areas, extending into precentral and postcentral cortex and pars triangularis, and also in structures of the posterior midline, such as precuneus and posterior cingulate. No areas of cortical thinning are observed in asymptomatic carriers, unlike those found in symptomatic people. Furthermore, structural changes are not limited to subjects with mutations in AD determinant genes, since they are also seen in the sporadic form of the disease. Some studies have shown that healthy subjects with evidence of initial deposits of Aβ present greater volume and thickness in AD related cortical regions. With AD progression, these areas suffer a progressive thinning of gray and also white matter reaching the atrophy observed in more extensive regions in the symptomatic stage.

It would be interesting to have more studies evaluating these compensatory mechanisms, to understand it better and, if possible, to add it to the current computational models evaluating AD’s pathology and possible treatments, as they seem to be present since the very early stages of the disease. “

The reader had difficulty following Table 1, particularly correlating the main criteria with the supporting criteria.  It is not clearly laid out. 

We have change table 1 in order to clarify information.

Reviewer 3 Report

«When does Alzheimer’s disease really start? The role of Biomarkers» seems to be a great contribution to the state-of-the-art. Nevertheless, a main point should be addressed.

In lines 194-197, authors mentioned a very recent work by Petrella and collaborators. The authors made a mistake and used the word «CASUAL» instead of «CAUSAL».

«A very recent work by Petrella and collaborators [85] developed a mathematical casual model of the dynamic biomarker cascade theory in AD, that might help to explain how these biomarkers interact and evolve over time, and that could potentially help patients, researchers and medical personnel.»

A great contribution of the present review to the state-of-the-art would be to delve deeper into this type of computational models, in terms of diagnosis and in the search for more specific and efficient treatments for AD. Reviews on Alzheimer’s pathology are very abundant and authors usually repeat ideas and concepts already reported in lots of articles. Then, the great novelty of the present review might be to focus on this type of study models. There is a lack of quantitative models explaining not only how biomarkers interact and evolve over time but simulating patient response to therapy prior to entry in clinical trials. 

Minor point:

What NIA-AA stands for should be indicated in the Abstract.

National Institute on Aging and the Alzheimer’s Association (NIA-AA)

Author Response

«When does Alzheimer’s disease really start? The role of Biomarkers» seems to be a great contribution to the state-of-the-art. Nevertheless, a main point should be addressed.

In lines 194-197, authors mentioned a very recent work by Petrella and collaborators. The authors made a mistake and used the word «CASUAL» instead of «CAUSAL».

«A very recent work by Petrella and collaborators [85] developed a mathematical casual model of the dynamic biomarker cascade theory in AD, that might help to explain how these biomarkers interact and evolve over time, and that could potentially help patients, researchers and medical personnel.»

We apologize with the reviewer for the mistake and we have corrected this error.

A great contribution of the present review to the state-of-the-art would be to delve deeper into this type of computational models, in terms of diagnosis and in the search for more specific and efficient treatments for AD. Reviews on Alzheimer’s pathology are very abundant and authors usually repeat ideas and concepts already reported in lots of articles. Then, the great novelty of the present review might be to focus on this type of study models. There is a lack of quantitative models explaining not only how biomarkers interact and evolve over time but simulating patient response to therapy prior to entry in clinical trials. 

We thank the reviewer for his/her deep revision of our manuscript. Thanks for this helpful and stimulating comment. We have added more details about the computational models available, focusing in its use for treatment prediction.

A very recent work by Petrella and collaborators [85] developed a mathematical causal model of the dynamic biomarker cascade theory in AD, that might help to explain how these biomarkers interact and evolve over time, and that could potentially help patients, researchers and medical personnel.

This is a great advancement in the knowledge of the disease but there is still a long way to go. Although, biomarkers could have a role in predicting whether a patient could convert from MCI to AD, there is not a consensus on which biomarkers could assume that role.

However, the scientific community’s efforts go beyond designing computational models to determine the behaviour of different biomarkers in the evolution of the disease. Models have been designed for many different aspects of the disease, such as a model based on the amyloid cascade hypothesis, showing the effects of pathological processes such as oxidative stress, inflammation or cerebrovascular disease in the kinetic of Aβ aggregation. Moreover, another model focused on synaptic loss and compensation by the reinforcement of the remaining connections and, more recently, Ding et al. (2018) designed a hybrid computational approach for a more accurate disease severity classification.

Nevertheless, as scientists started to better understand AD’s pathophysiology, the biggest challenge became designing computational models capable of predicting the efficacy of a specific treatment. To reach this objective, models have been created analysing potential treatments. Anastasio (2013) incorporated the role of estrogens in Aβ regulation into a model that can generate therapeutic predictions and the possible benefits of this therapy. This model showed that estrogen could reduce Aβ and that non-steroidal anti-inflammatory drugs could provide a small additional benefit.

Furthermore, immunotherapy, probably the most promising treatment for AD at this moment, was also analysed by computational models. Diem et al (2016), have incorporated this therapy’s possible complications into their model and concluded that a failure in periarterial drainage seems to be an important mechanism. Another computer simulation model pointed out that immunotherapy against Aβ might not be effective unless it’s used during early stages of AD or combined with other therapies. However, a more recent model simulated the differential impact of Aβ oligomers on glutamate and nicotinic neurotransmission while under different treatments, including a passive vaccination with the monoclonal antibody solanezumab, the use of the beta-secretase inhibitor verubecestat and of the gamma-secretase inhibitor semagacestat. They predicted a cognitive worsening in people with low Aβ baseline and an improvement in those with moderate to high Aβ levels.

Computational models analyzing neurotransmitters have also been created. One such model has been implemented using preclinical data available on receptor pharmacology of cholinergic and catecholamine neurotransmitters and clinical data, to predict the effects of memantine, an N-Methyl-D-aspartic acid (NMDA) inhibitor, in different phases of AD pathology.

Lastly, Stefanovski et al (2019) created a computational multi-scale brain model, using The Virtual Brain Platform, and including PET and electroencephalogram, to simulate regional neural activity and hyperexcitability in AD and how it relates to Aβ. This model reveals a potential functional reversibility of large-scale alterations in AD after memantine treatment.

Minor point:

What NIA-AA stands for should be indicated in the Abstract.

National Institute on Aging and the Alzheimer’s Association (NIA-AA)

We have added the meaning of NIA-AA in the Abstract Section.

Round 2

Reviewer 2 Report

Well done.

Reviewer 3 Report

The review has been improved significantly by description of appropriate works performed on mathematical models in terms of diagnosis and in the search for more specific and efficient treatments for AD.